# Reducing the False Negative Rate in Deep Learning Based Network Intrusion Detection Systems

Jovana Mijalkovic [†] and Angelo Spognardi *,[†]

Department of Computer Science, Sapienza University, 00198 Rome, Italy;
mijalkovic.1908929@studenti.uniroma1.it
* Correspondence: spognardi@di.uniroma1.it
† These authors contributed equally to this work.

**Abstract:** Network Intrusion Detection Systems (NIDS) represent a crucial component in the security of a system, and their role is to continuously monitor the network and alert the user of any suspicious activity or event. In recent years, the complexity of networks has been rapidly increasing and network intrusions have become more frequent and less detectable. The increase in complexity pushed researchers to boost NIDS effectiveness by introducing machine learning (ML) and deep learning (DL) techniques. However, even with the addition of ML and DL, some issues still need to be addressed: high false negative rates and low attack predictability for minority classes. Aim of the study was to address these problems that have not been adequately addressed in the literature. Firstly, we have built a deep learning model for network intrusion detection that would be able to perform both binary and multiclass classification of network traffic. The goal of this base model was to achieve at least the same, if not better, performance than the models observed in the state-of-the-art research. Then, we proposed an effective refinement strategy and generated several models for lowering the FNR and increasing the predictability for the minority classes. The obtained results proved that using the proper parameters is possible to achieve a satisfying trade-off between FNR, accuracy, and detection of the minority classes.

**Keywords:** NIDS; deep learning; false negative rate; machine learning; artificial neural network

## 1. Introduction

Since the introduction of the first Intrusion Detection Systems, one of the biggest challenges they faced was a high False Positive Rate (FPR) which means that they generate many alerts for non-threatening situations. Security analysts have a massive amount of threats to analyze, which can result in some severe attacks being ignored or overlooked [1]. Another challenge was the False Negative Rate (FNR), which was still not low enough. A high FNR presents an even bigger problem than a high FPR because it is more dangerous to falsely classify an attack as regular network traffic than vice versa. Because of the constant technological improvements and network changes, new and more sophisticated types of attacks emerge, creating the need for continuous improvement of Intrusion Detection Systems.

One way of improving IDSs, on which the researchers have been working in the last years, is using machine learning techniques to reduce the FPR and FNR and improve general detection capabilities [2]. A good example can be found in [3], where the authors developed a prototype IDS which aimed to detect data anomalies by using the k-means algorithm implemented in Sparks MLib. The reason behind using ML algorithms is that they can analyze massive amounts of data and gather any information which can then be used to enhance the capabilities of IDSs [1]. Another reason for using ML algorithms is that they are not domain-dependent and are very flexible- functional for multiple problems [4].

Researchers identified two primary issues in the literature regarding the already-existing deep learning models used for IDS [5]. The first issue is that some have low

detection accuracy, especially when dealing with unbalanced data [6]. Most of the research that focuses on the problem of machine learning and deep learning intrusion detection systems uses the same publicly available datasets. After analyzing the extensive work available on this topic, it emerges that some classes had meager detection rates when it comes to multiclass classification, as will be presented in Section 3. The second issue is that some models have somewhat high accuracy but also high False Positive and False Negative Rates, which can lead to lower detection efficiency and weaken the network security [7,8]. Aside from these problems, the datasets used in some research are very aged and might not reflect the modern-day network traffic, so the question arises: can these Intrusion Detection Systems detect modern-day attacks? Moreover, how much can NIDS based on Deep Neural Networks reduce the quite dangerous false negatives?

The objective of this research is tackle the above mentioned problems, and propose a robust solution to improve the detection quality of Network Intrusion Detection Systems using deep learning techniques, namely artificial neural networks. More specifically, the idea is to lower the FNR and FPR and increase the attack predictability of the less represented attack types. We state that, from a security perspective, we could tolerate a slight increase in the FPR if this is a price for nullifying the FNR because it is more dangerous to wrongly classify an attack as benign traffic than the other way around.

We start building a deep neural network for network intrusion detection purposes. The deep neural network will be fed using two different datasets for binary and multiclass network traffic classification. The models will be able to differentiate between regular network traffic and attacks, as well as between different categories of attacks. We then propose a strategy to lower the False Negative Rate of the models by doing various experiments with different methods to reduce the FNR while keeping the False Positive Rate low and the other metrics such as accuracy, precision, and recall high. The strategy we use can be summarized in three steps: modifying the distribution of the training and testing datasets, reducing the number of dataset features, and using class weights. For our purposes, we used two different datasets, NSL-KDD [9] and UNSW-NB15 [10]. The idea was to train the neural network models using an older and a more recent dataset and, in that way, include a more extensive range of network attacks that the network will be able to detect.

The rest of the paper is structured as follows: the next section provides the theoretical background, with the introduction of the building blocks of our research. Section 3 describes the used dataset and surveys the literature's main results related to the deep learning approach for NIDS. This section will show the reduced concern of the related work about lowering false positives and false negatives. Section 4 provides the details of our approach, while Sections 5 and 6 report the descriptions and the results of the experimental campaign. Finally, Section 7 concludes the paper with some overall observations and some future investigation directions.

## 2. Theoretical Background

In this section we present the fundamental elements to have a reference background, namely Intrusion Detection Systems and artificial neural networks for deep learning.

### 2.1. Intrusion Detection Systems

An Intrusion Detection System is a software application or a device that monitors network traffic and computer usage intending to detect any suspicious actions that go against regular or expected use of the system, for example, a harmful activity or a policy breach, in order to allow for system security to be maintained. Once the system detects such actions, it alerts the user and collects information on the suspicious activity [11].

Network Intrusion Detection Systems are designed to protect the whole network, not just a single host. NIDS are placed in strategic positions, for example, at the edge of the network, where a network is most vulnerable to attacks [12]. NIDS analyze inbound and outbound traffic to see if it fits the expected average behavior or matches known

attack patterns. One positive aspect of this type of IDS is that it can be tough to detect its presence in a system, which means that, usually, an attacker will not even realize that NIDS is scrutinizing his actions. On the other hand, one negative aspect is that this type of IDS analyzes enormous amounts of traffic, which leaves space for making mistakes and generating an excess of false positives, or even some false negatives [13]. To avoid this, they need more fine-tuning done by the administrator to ensure that they are working correctly and not missing anything that might be crucial to the network's security.

IDS need to know how to differentiate between suspicious and regular behavior. For this purpose, there are different methods that they can use. The two main detection approaches are called *signature-based* detection and *anomaly-based* detection [11]. The signature-based approach, also known as *knowledge-based* or *definition-based*, uses a database of known vulnerabilities, signatures (byte combinations), or attack patterns. It identifies attacks by comparing them to this database [12]. The underlying idea is to have a database of anomalies recognized as attacks so that IDS can detect, promptly alert, and possibly avoid the same (or similar) events in that database.

The anomaly-based approach, also known as the behavior-based, focuses on identifying instances of malicious behavior, or in other words, system or network activity that does not fit the expected behavior. These instances are called outliers, and once the IDS detects an outlier, it is supposed to warn the administrator about it. Unlike signature-based IDS, anomaly-based IDS can detect and alert the administrator when they discover a suspicious behavior unknown to them. Instead of searching through a database of known attacks, anomaly-based IDS use machine learning to train their detection system to recognize a normalized baseline, which typically represents how the system behaves. Once the baseline is determined, all the activity is compared to this baseline to see what stands out from the typical network behavior [14].

### 2.2. Artificial Neural Networks for Deep Learning

Machine Learning (ML) is a specific branch of computer science and artificial intelligence (AI) that focuses on using existing data and algorithms to mimic how people think, learn and make decisions while gradually improving the accuracy of the decision-making process and its results [15]. ML algorithms build a mathematical model using sample data, also known as training data, aiming to make decisions that they are not explicitly programmed to make [16].

Artificial neural networks (ANN), usually only called neural networks (NN), are computing systems that contain a group of artificial neurons used to process data and information. The architecture of the ANNs, and the idea behind building them, is based on the biological neural networks found in human brains. The artificial neurons (nodes) are a collection of connected units loosely modeled on the human brain's neurons. The idea is that these neurons should replicate how the human brain works. At its core, the neuron is a mathematical function, which takes an input, does a calculation and transformation on it, and gives an output.

Deep learning is essentially a subfield of machine learning, and it represents a particular case of an artificial neural network having more than one hidden layer. As previously mentioned, these types of neural networks aim to simulate the human brain and learn from large amounts of data [17]. The idea behind adding additional hidden layers is to increase accuracy and optimize the model. The difference between deep learning and machine learning is in the type of data they use and the methods they use to learn. Machine learning usually uses structured, labeled data to make predictions. Even if the data are not structured, they usually go through the data preparation phase to be organized in a way that the learning model can use. On the other hand, deep learning can use data that are not structured, such as images and text, which means that these algorithms can shorten the processing phase or even remove it altogether [18].

In recent years, machine learning methods have been extensively used to build efficient network intrusion detection systems [1]. The use of machine learning methods has

significantly impacted and improved the detection accuracy of these intrusion detection systems. However, there are still some downsides and limitations to using shallow machine learning methods. In particular, they still require a high level of human interaction and a significant amount of expert knowledge to process data and identify patterns [19], making them expensive, time-consuming, and unsuitable for high-dimensional learning with a large amount of data. Another negative side of using shallow machine learning techniques is that their learning efficiency decreases as the network complexity increases. When there are many multi-type variables, logistic regression can underfit with low accuracy, decision trees can overfit, and Support Vector Machines are inefficient, mainly when dealing with large amounts of data [20].

To address these limitations, researchers have identified Deep Learning as a valid alternative to shallow learning techniques in the above mentioned situations. Advantages of DL over ML are, for example, automatic feature learning and flexible adaptation to novel problems, making it easier to work with big data [20].

## 3. Related Work

Deep Learning for NIDS is an emerging topic that has generated a new research branch. There have been many novel approaches proposed by authors, such as in [21], where the authors have proposed a modified bio-inspired algorithm, which is the Grey Wolf Optimization algorithm (GWO), that enhances the efficacy of the IDS in detecting both normal and anomalous traffic in the network. Another example is [22], where the researchers analyzed the evolutionary sparse convolution network (ESCNN) intrusion and threat activities in the Internet of things (IoT) with the goal to improve the overall attack detection accuracy with a minimum false alarm rate. In this section, we report our analysis of the main proposals found in the literature. The discussion will include an analysis of the deep learning methods and the datasets used, the models produced, and the results obtained for each research paper. We separated the different research proposals according to the dataset they adopted to build their deep neural network models, namely the NSL-KDD [9] and NSW-NB15 [10] datasets. The selection process of the related literature was based on the following criteria:

1. Usage of the NSL-KDD and UNSW-NB15 datasets
2. Being relevant to Network Intrusion Detection Systems
3. Usage of deep learning algorithms

### 3.1. Datasets for Training Deep Learning Based NIDS

Training machine learning algorithms requires huge amounts of data, and the quality of these data is crucial. Since most problems are very dependent on the type and the quality of data, high quality datasets need to be used. Both NSL-KDD and UNSW-NB15 datasets have been used in many previous IDS researches, as described in the following Sections 3.2 and 3.3.

The original researchers produced the NSL-KDD dataset to try to solve the shortcomings and problems of the KDD Cup 99 dataset, once the most widely used dataset for the evaluation of anomaly detection methods, prepared by Stolfo et al. [23]. The KDD Cup 99 dataset's biggest problem was biased results due to redundant and duplicate records. The NSL-KDD dataset consists of selected records from the complete KDD Cup 99 dataset. This new dataset removes the identical records, resulting in around 78% of the training dataset records and around 75% of the test dataset records [24]. Moreover, the number of selected records from each difficulty level group is inversely proportional to the percentage of the records in the original KDD Cup 99 dataset [25]. The NSL-KDD dataset contains both regular traffic and traffic representing network attacks, so all the data in the dataset are labeled as either normal or attack.

The NSL-KDD dataset is divided into four datasets: KDDTest+, KDDTrain+, KDDTest-21, and KDDTrain+20%, where the latter are subsets of the former two, respectively. The KDDTest-21 is a subset of the KDDTest+ and excludes the most challenging records. Simi-

larly, the KDDTrain+20% is a subset of the KDDTrain+ and contains 20% of the records in the entire training dataset [26].

The training dataset consists of 21 different attack types, while the testing dataset has 39 different types of attacks. The attack types in the training dataset are considered known attacks, while the testing dataset consists of the known attacks, plus the additional, novel attacks. The attacks are grouped into DoS, Probe attacks, U2R, and R2L. More than half of the records are regular traffic, while the distribution of the R2L and U2R attacks is low. On the other hand, a lower distribution corresponds to real-life internet traffic attacks, where these types of traffic are very rarely seen [26]. The dataset includes a total of 43 features. The first 41 are related to the traffic input and are categorized into three types: basic features, content-based features, and traffic-based features.

The distribution of the above mentioned attack types is skewed and the breakdown of the data distribution can be seen in Table 1. More than half of the records are normal traffic, while the distribution of the R2L and U2R attacks is low.

**Table 1.** NSL-KDD record distribution.

|  | Total | Normal | DoS | Probe | U2R | R2L |
|---|---|---|---|---|---|---|
| KDDTrain+ | 125,973 | 67,343 (53%) | 45,927 (37%) | 11,656 (9.11%) | 52 (0.04%) | 995 (0.85%) |
| KDDTest+ | 22,544 | 9711 (43%) | 7458 (33%) | 2421 (11%) | 200 (0.9%) | 2654 (12.1%) |

The UNSW-NB15 dataset is a relatively new network dataset, released in 2015 and used in developing NIDS models [10]. The authors reported several main reasons for making this new dataset. They wrote, in fact, that available datasets were too old, did not reflect modern network traffic, and did not include some essential modern-day attacks. The original dataset consists of 2,540,044 records, which can be classified as regular traffic and network attacks. The authors have also made two smaller subsets, the training, and testing subsets, consisting of 175,341 and 82,332 records, respectively. The original dataset distinguishes a total of 49 features, and the authors arranged 35 in four categories: flow, basic, content, and time features. These 35 features hold the integrated gathered information about the network traffic. The following 12 features are additional and grouped into two groups based on their nature and purpose. The first group contains features 36–40, considered general-purpose features, while the remaining 41–47 are considered connection features [10]. Each of the general-purpose features has its purpose from the defense point of view, while the connection features give information in different connection scenarios. The remaining two features, 48 and 49, are the label features, and they represent the attack category and the traffic label, which shows whether the record is regular traffic or an attack, respectively.

Similarly to the NSL-KDD dataset, the UNSW-NB15 dataset is also very unbalanced. The breakdown of the data distribution can be seen in Table 2.

**Table 2.** UNSW-NB15 record distribution. Normal traffic accounts for 87% of the total, while some attacks are <0.00005% (i.e., Shellcode and Worms).

| Normal | Fuzzers | Analysis | Backdoors | DoS |
|---|---|---|---|---|
| 2,218,761 | 24,262 | 2677 | 2329 | 16,353 |
| Exploits | Generic | Reconnaissance | Shellcode | Worms |
| 44,525 | 215,481 | 13,987 | 1511 | 174 |
|  |  | Total |  |  |
|  |  | 2,540,044 |  |  |

*3.2. Related Research Using the NSL-KDD Dataset*

This section surveys the research papers which used the NSL-KDD dataset for training and testing of the model.

Jia et al. [27] considered the two datasets, KDD Cup 99 and NSL-KDD, and proposed a network intrusion detection system based on a deep neural network with four hidden layers. Each hidden layer has 100 neurons and uses the ReLU activation function. The output layer is fully connected and uses the softmax activation function. The authors have built a multiclass classifier with the final aim to increase the model's accuracy. In the end, they have obtained an accuracy of >98% on all the classes except the U2R and R2L attacks. The authors claimed that the main reason is the severely unbalanced nature of the datasets, since there are too few records for these classes. We can observe two main downsides of this research: it uses a very old dataset (KDD Cup 99), and the two used datasets are very similar. This last point could mean that, even though this model performs well on these datasets, it might not perform as well when detecting in a real network environment.

Vinayakumar et al. [28] proposed an intrusion detection system based on a hybrid scalable deep neural network. They tested their model using six different datasets: KDD Cup 99, NSL-KDD, Kyoto, UNSW-NB15, WSN-DS, and CICIDS 2017. The proposed model consists of an artificial neural network with five hidden layers using the ReLU activation function. Each hidden layer has a different number of neurons ranging from 1024 in the first hidden layer to 128 in the last. The authors evaluated both binary and multiclass classification, obtaining broadly varied results. Depending on the dataset used, the proposed models obtained the best accuracy for the KDD Cup 99 and the WSN-DS, and the worst for the NSL-KDD and the UNSW-NB15. The authors' main goal was to develop a flexible model that can detect and classify different kinds of attacks, which is why they used multiple datasets. The downsides of the proposed approach are that the obtained model is very complex and has a lower detection rate for some of the classes.

Another research on this topic was done by Yin et al. [29]. Their study proposed a network intrusion detection system based on a Recurrent Neural Network (RNN) model. The dataset used in this research is the NSL-KDD dataset, and the authors have trained an RNN model to do both binary and multiclass classification. The idea behind the study was to build a model that will achieve higher performance in attack classification than the models using the more traditional machine learning algorithms, such as Random Forest, Support Vector Machine, etc. After the data preparation phase, the dataset used to train the model consisted of 122 features, while the final model consisted of 80 hidden nodes. The accuracy results obtained when testing the model were: 83.28% for binary classification and 81.29% for multiclass classification. The authors state that these results are better than those of other machine learning algorithms. Some downsides of this approach are that the detection rates for the R2L and U2R classes are still low, and the model's performance is lower than other deep learning IDS models.

Potluri et al. [30] propose a DNN architecture of a feed-forward network where each hidden layer is an auto-encoder, trained with the NSL-KDD dataset. Using auto-encoders as hidden layers allows the training process to be done one layer at a time. The network has three hidden layers: the first two are auto-encoders, with 20 and 10 neurons, respectively; the third layer uses the softmax activation function and has five neurons. The first two layers are used in the pre-training process: they perform a feature extraction phase and reduce the number of features used by the DNN first to 20 and in the end to 10. The third hidden layer selects five features out of 10 as a fine-tuning phase. The experiments considered binary and multiclass classification: the detection accuracy for the binary classification is high (>96%). In contrast, the detection accuracy for multiclass classification varied considerably: it was satisfactory (>89%) for DoS, Probe, and regular traffic and low for U2R and R2L. Similar to other research papers mentioned, the low detection accuracy for some classes is a downside of this model.

Kasongo et al. [31] also proposed a network intrusion detection system based on a feed-forward deep neural network using the NSL-KDD dataset. The goal of the research

was to build a model that would perform better, meaning it would have a higher detection accuracy than the existing machine learning models used for intrusion detection. The authors divided the original training dataset into two subsets: one for training and one for the evaluation after the training process. The initial test dataset was used to test the performance of the model. The experiment included binary and multiclass classification in two scenarios: the first used all 41 features of the dataset, and the second used a reduced number of features (21 features) extracted during the feature selection phase. The model with all the features showed a detection accuracy of 86.76% for binary and 86.62% for multiclass classification. On the other hand, when using the reduced number of features, the detection accuracy was 87.74% for binary and 86.19% for multiclass classification. Among the downsides of this model were lower detection rates for R2L and U2R classes and lower accuracy compared to other deep learning models used for intrusion detection.

The research paper by Shone et al. [19] also focuses on building a network intrusion detection system based on a deep learning model using the KDD Cup 99 and the NSL-KDD datasets. The proposed model is constructed by stacking non-symmetric deep auto-encoders and combining them with the Random Forest classification algorithm. One of the research purposes is to develop a technique for unsupervised feature learning, and the authors have done this by using another non-symmetric deep auto-encoder. The authors proposed two classifications: a 5-class classification for both datasets and a 13-class classification for the NSL-KDD dataset. The average detection accuracy for the 5-class classification was 97.85% with the KDD Cup 99 dataset, and 85.42% for the NSL-KDD dataset, while achieving 89.22% for the 13-class classification with the NSL-KDD dataset. The downside of this model is that it has low detection accuracy for classes with a lower number of records.

The research paper by Fu et al. [32] proposes a deep learning model for network intrusion detection with the goal to address the issue of low detection accuracy in imbalanced datasets. The authors have used the NSL-KDD dataset for the training and testing of the model. The model combines an attention mechanism and the bidirectional long short-term memory (Bi-LSTM) network, by first extracting sequence features of data traffic through a convolutional neural network (CNN) network, then reassigning the weights of each channel through the attention mechanism, and finally using Bi-LSTM to learn the network of sequence features. This paper employs the method of adaptive synthetic sampling (ADASYN) for sample expansion of minority class samples, in order to address data imbalance issues. The experiments included both binary and multiclass classification and the accuracy and F1 score of the proposed network model reached 90.73% and 89.65% on the KDDTest+ test set, respectively.

### 3.3. Related Research Using the UNSW-NB15 Dataset

This section discusses the research papers which used the UNSW-NB15 dataset for training and testing of the model.

In the research by Kanimozhi et al. [33], the authors proposed a network intrusion detection system based on an artificial neural network, trained and tested on the UNSW-NB15 dataset. The authors used deep learning in combination with other machine learning algorithms to extract the most relevant features of the dataset and use them for training the model. The goal was to increase the detection accuracy and decrease the False Alarm Rate. During the feature extraction phase, the authors used a combination of the Random Forest and the Decision Tree algorithms for feature extraction. In the end, they selected four features out of 45 in the original dataset. The authors have decided to do only binary classification, meaning that the model will only classify a record as an attack or regular traffic. The accuracy obtained in the testing phase was 89%, which is still lower than the accuracy of other proposals with deep learning approaches.

Mahalakshmi et al. [34] have implemented an intrusion detection system based on a convolutional neural network (CNN). The goal was to make a model that would overtake the existing machine learning models used for intrusion detection concerning detection accuracy. The proposed algorithm is a CNN used for binary classification, with an accuracy of 93.5%.

The research done by Al-Zewairi et al. [35] uses the whole dataset, with all 2,540,044 records, instead of the separate training and testing datasets prepared by the authors of the UNSW-NB15 dataset. The proposed model is a deep artificial neural network consisting of five hidden layers and a total of 50 neurons. The neural network is feed-forward and uses backpropagation and stochastic gradient descent. The research aimed to find the optimal network hyperparameters to achieve the best performance for binary classification. The authors conducted experiments to find the best activation function for their model and the optimal features to be used for training. The activation function that proved optimal for this research was the rectifier function without the dropout method. The second experiment regarding the optimal features showed that using the top 20% features, which were selected during feature extraction, gave the best results. After testing the proposed model, the evaluation showed high accuracy (98.99%) and a low false alarm rate (0.56%).

We can note that few researchers, from the ones mentioned in this section, included the FPR and FNR as an evaluation metric in their research. However, most of them focused on calculating the accuracy. The main problem with this approach is that the datasets used are significantly unbalanced. Therefore the accuracy is not a good metric because it does not distinguish between the records of different classes that were correctly classified. With this concern in mind, in this paper we propose to focus on lowering the FNR and increasing the predictability for the minority classes.

### 3.4. Summary and Comparison of the Related Research

A summary and comparison of all of the surveyed research papers are in Table 3. We can observe that only half of the authors included the FPR and FNR as an evaluation metric in their research since most of them focused on improving the accuracy. Moreover, only two of the authors that considered the False Rates also proposed a multiclass classification.

The main problem of focusing on the accuracy metric is that the datasets used are significantly unbalanced. Therefore the accuracy is not a good metric because it does not distinguish between the records of different classes that were correctly classified. Thus, in the next we focus on a strategy to improve the FNR and FPR, while improving the detection of the less represented attack classes. In order to provide a better overview and the possibility to compare the related work with the results which were achieved by the model proposed in this research, we have included a brief summary of the proposed model as the last row in Table 3.

**Table 3.** Summary and comparison of related works.

| Researchers | Year | Dataset(s) | Algorithm(s) | Classification Type | Accuracy | FPR and FNR |
|---|---|---|---|---|---|---|
| Jia et al. [27] | 2019 | KDD Cup 99 and NSL-KDD | Deep neural network | Multiclass | >98% on all classes except U2R and R2L | FNR = 0.5%, FPR = 0.3% |
| Vinayakumar et al. [28] | 2019 | KDDp Cup 99, NSL-KDD, Kyoto, UNSW-NB15, WSN-DS and CICIDS 2017 | Deep neural network | Binary and multiclass | Big variations between datasets | Big variations between datasets |
| Yin et al. [29] | 2017 | NSL-KDD | Recurrent neural network | Binary and multiclass | 83.28% for binary and 81.29% for multiclass | N/A |
| Potluri et al. [30] | 2016 | NSL-KDD | Deep neural network with auto-encoders as hidden layers | Binary and multiclass | >96% for binary; >89% for multiclass | N/A |
| Kasongo et al. [31] | 2019 | NSL-KDD | Deep neural network | Binary and multiclass | All features: 86.76% (binary), 86.62% (multiclass); 21 features: 87.74% (binary) and 86.19% (multiclass) | N/A |

**Table 3.** *Cont.*

| Researchers | Year | Dataset(s) | Algorithm(s) | Classification Type | Accuracy | FPR and FNR |
|---|---|---|---|---|---|---|
| Kanimozhi et al. [33] | 2019 | UNSW-NB15 | Deep neural network | Binary | 89% | FNR = 15% |
| Mahalakshmi et al. [34] | 2021 | UNSW-NB15 | Convolutional neural network | Binary | 93.5% | N/A |
| Shone et al. [19] | 2018 | KDD Cup 99 and NSL-KDD | Stacked non-symmetric deep auto-encoder network with Random Forest classification algorithm | Multiclass (5 and 13 classes) | 97.85% (5-class KDD Cup 99); 85.42% (5-class NSL-KDD) and 89.22% (13-class NSL-KDD) | Only FPR considered, big variations between experiments (from 2.15% to 14.58%) |
| Al-Zewairi et al. [35] | 2017 | UNSW-NB15 | Deep neural network | Binary | 98.99% | FPR = 0.56% |
| Fu et al. [32] | 2022 | NSL-KDD | Deep neural network | Binary and multiclass | 90.73% | Lowest FPR for U2R class (1.73%), highest for Normal class (13.44%) |
| Mijalkovic J., Spognardi A. (proposed model) | 2022 | NSL-KDD and UNSW-NB15 | Deep neural network | Binary and multiclass | >99% for NSL-KDD and >97% for UNSW-NB15 | Lowest FNR = 0.049%; lowest FPR = 0.33% |

## 4. Materials and Methods

In this section, we present the strategy we propose to achieve our research goals, while in the next Section 5, we report the experimental campaign that confirms our approach.

Our strategy to reduce FNR and FPR and increase the detection of low-represented attack categories consists of three points, as depicted in Figure 1. The first point, *distribution alteration*, refers to the idea of altering the distribution of the original datasets. The rationale is that the split proposed by the original dataset's authors is sub-optimal, limiting the final accuracy of the trained model. Our idea is that by reshuffling the datasets, it is possible to improve the detection rate of most of the attack categories.

The second point, *feature reduction*, is the canonical approach of reducing the number of features [36], selecting the more suitable for the primary goal.

The final point, *class weight*, refers to the idea of altering the importance of the different categories of the data samples used in the network. The rationale is that we can reduce the number of false negatives and improve the detection of the less common attacks at the price of a low increase in the number of false positives.

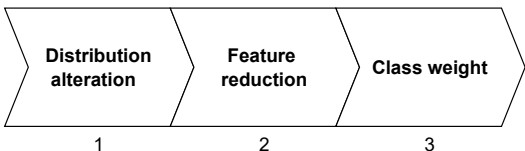

**Figure 1.** An overview of the proposed strategy.

In the Experiment section (Section 5), we reported and evaluated all the intermediate results to show the improvement introduced by each of the points of our strategy.

### 4.1. Strategy Implementation

Figure 2 shows the details of the phases we took to construct and evaluate the generated models. In the following, we give an overall description of each phase, and the details of the data preparation and model architecture in Sections 4.3 and 4.2, respectively.

The first step was to collect the data. As mentioned in Section 3.1, we selected NSL-KDD and UNSW-NB15 to have two different datasets considered among the most suitable

for our research. Since both datasets are divided into smaller datasets, the following were chosen for our research: KDDTrain+, KDDTest+, and the full UNSW-NB15 dataset, which we split into 4 CSV files.

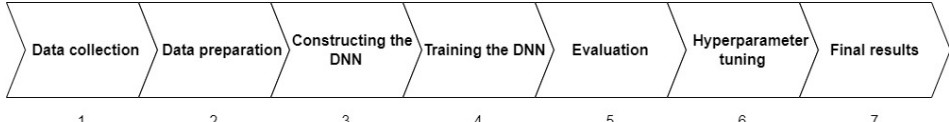

**Figure 2.** An overview of the steps taken to build and evaluate the Deep Learning model.

The next step is to prepare the datasets so they can be ready to be used for training our model. In this phase, we processed the datasets by removing missing and redundant values, normalizing the numerical data, and encoding the categorical data into numerical. Section 4.2 gives a detailed explanation of this step.

The third phase is constructing the deep neural network used in the research and setting all of its parameters. A detailed explanation of the architecture and the parameters chosen for the model is given in Section 4.3.

The fourth step is essential in deep learning and consists of training the neural network since the dataset is used to train the model and enhance its ability to make predictions.

After the training of the model, the fifth step is to evaluate the model to see how it performs. The testing datasets are used in this step, in order to see how well the model will perform on the data that it has never seen before.

After the evaluation process, the next step is to tune the hyperparameters to see if it would be possible to improve the learning process and achieve better results. The hyperparameters are the parameters used to control the learning process, as opposed to other parameters, such as node weights, whose values come from training. Some of the parameters modified in this phase to obtain better results are, for example, the number of epochs and the learning rate. We detail all the values and the obtained results of this step in Section 4.3.

The final step is the prediction step, in which we achieve the final results of the model. In this step, we conclude how well the model performed and if it reached the experiment's goal. The predictions of each experiment and the evaluation of their results are in Sections 5 and 6, respectively.

*4.2. Data Preparation*

Preparing the data is a crucial step and can significantly impact the model's learning process. If we do not give appropriate input to the model, it might not give us the result that we want to obtain. As mentioned earlier, we have two datasets used in this research, the NSL-KDD and the UNSW-NB15 datasets. Both of these datasets need to be processed, and since they have a similar structure, we used the same preparation process.

4.2.1. Preparation of the NSL-KDD Dataset

As a starting point in the data preparation of the NSL-KDD dataset, we have two subsets of data already divided by the authors, the KDDTrain+ and the KDDTest+. These subsets have 43 features, while the KDDTrain+ subset has 125,793 records and the KDDTest+ subset 22,544. We processed and verified that both subsets do not contain any missing values. Therefore, we could proceed with doing the rest of the data preparation on the subsets as they are.

The goal of multiclass classification is to correctly classify records that represent a network attack as the attack category they belong to. Therefore, it is necessary to change the label for every record from the attack type to the class to which that attack type belongs. This step is repeated for both subsets. For the model to learn from this data, we need to transform it into numerical values. For this transformation, we employed one-hot encoding. One-hot encoding is a technique used for categorical features where no ordinal

relationship exists. Therefore it is not enough to just do integer encoding (assign each category an integer). One-hot encoding creates new binary columns for each possible unique categorical feature value. In other words, it converts the categorical data into a binary vector representation. We applied one-hot encoding to training and test subsets specifically for the following features: protocol_type, service, and flag. Ultimately, we removed the original categorical columns and obtained a dataset with 124 columns.

The next step was the encoding of the label. For binary classification, the 'normal' value was represented by a 0, while all the others 'abnormal' were given the value 1. For multiclass classification, again, the 'normal' value was given the value 0, and the rest of the values were integer encoded. The multiclass values range from 0 to 4. This was done for both subsets.

The next step was to strip the label and attack category columns from the train and test datasets, building the effective subsets used to generate the model. The combination of the original subset with the label column is used for the binary classification, while the combination with the attack category column is used for the multiclass classification. Thus, we divided the training and the testing subsets into 6 subsets: $train_f, train_\ell, train_c, test_f, test_\ell,$ and $test_c$. The subsets $train_f$ and $test_f$ contain all the columns with the features of the original training and testing datasets except for label and attack category columns: they will be given to the model as the input. The label column for training and testing for binary classification went in $train_\ell$ and $test_\ell$, respectively, while the attack category column went in $train_c$ and $test_c$.

The last preparation step was to normalize the data in the $train_f$ and $test_f$ subsets using the min-max method. For every feature, the minimum value is changed to 0, the maximum value is changed to 1, and every other value is transformed into a decimal value between 0 and 1 using the following formula $\frac{value-min}{max-min}$. The final subsets used, $train_f$ and $test_f$, now contain 123 columns each, and all the data is encoded into numerical values and normalized.

### 4.2.2. Preparation of the UNSW-NB15 Dataset

Unlike the NSL-KDD dataset, we opted to use the original full UNSW-NB15 dataset, which contains 2,540,044 records, instead of using the two subsets pre-divided by the authors. The authors have provided four separate CSV files which contain the records of this dataset. The first step was to load all four CSV files and merge them into one dataset.

The next step was to check if there were any duplicate records and remove them. The removal of the duplicates is essential to avoid having the same records in the training and testing subsets because the testing subset should contain only the records that were not previously seen by the neural network. During this phase, we removed 480,625 duplicate records.

The next step was to check if the dataset contains any missing values. Three features contained missing values: 'ct_flw_http_mthd', 'is_ftp_login' and 'ct_ftp_cmd'. The missing values were then replaced with '0'. It has been noted that the dataset contains the value '–' for the feature 'service' in a significant number of records, so this value was renamed as 'undefined' to give more meaning to it. Then, we removed the columns 'srcip' and 'dstip'. We also fixed some white-space inconsistencies among records with the same values and other minor typos (i.e., 'Backdoors' instead of 'Backdoor' in the 'attack_cat' field).

We repeated the one-hot encoding for the whole dataset, changing the categorical features 'proto', 'service', and 'state'. At the end of this process, the dataset contained 202 columns.

While the column 'label' used for binary classification already contained 0 for regular traffic and 1 for abnormal, the 'attack category' required an encoding for the multiclass classification. Thus, in the next step, we encoded with a 0, the 'normal' (no-attack) value, and assigned values from 1 to 9 to the other attack categories.

The next step was to split the dataset into training and testing subsets. The training subset was a random sample with 80% of the original records, while the testing subset contained a random sample with 20%.

As for the NSL-KDD dataset, we separated the feature data columns ($train_f$ and $test_f$) from the label ($train_\ell$ and $test_\ell$) and attack category ($train_c$ and $test_c$) columns.

As for the NSL-KDD dataset, the final step was the normalization of the numerical variables of the $train_f$ and $test f$ subsets of the features with the min-max normalization method. In the end, these subsets contain 200 columns.

### 4.3. Model Architecture

After the data preparation phase, we started training the deep neural network. We adopted the same model architecture for both datasets to evaluate which would perform better. Different activation functions are used for different layers of the neural network. We differentiated the model for the binary classification and the one for multiclass classification, changing the number of nodes in the output layer and the activation function for the output layer. The hyperparameters related to the training algorithm are:

- Batch size. This is a training parameter that indicates the number of records passed and processed by the algorithm before updating the model.
- Number of epochs. This is also a training parameter which indicates the number of passes done through the complete training dataset.
- Optimizer. Optimizer is an algorithm, or a method, which is used to change the attributes of the network such as weights and learning rate in order to reduce the loss. The most used optimizers, among the others, are gradient descent, stochastic gradient descent, adagrad, and adaptive moment estimation (Adam) [37]. The optimizer used for the model is stochastic gradient descent (SGD) with Nesterov momentum.
- Momentum. This parameter is used to help predict the direction of the next step, based on the previous steps. It is used to prevent oscillations. The usual choice is a number between 0.5 and 0.9.
- Learning rate. The learning rate is a parameter which controls the speed at which the neural network learns. It is usually a small positive value in range between 0.0 and 1.0. This parameter controls how much we should change the model in order to respond to the estimated error each time the weights of the model are updated [38].
- Loss function. The loss function in a neural network is used to calculate the difference between the expected output and the output that was generated by the model. This function allows acquiring the gradients that the algorithm will use to update the neural network's weights. The loss function used for this model for binary classification is the binary cross-entropy loss function. On the other hand, we used a sparse categorical cross-entropy loss function for multiclass classification.

At the end of our experiments, the final values chosen for the training are provided in Table 4. These final values were reached after a process of manual hyperparameter tuning which included a series of trials with different values. The number of epochs shown in Table 4 indicates the maximum number of epochs, but Early Stopping is used in the experiments in order to prevent overfitting.

The neural network used for the experiment is a feed-forward neural network, which means that the connections between the nodes do not form any cycles and the data in the network moves only forward from the input nodes, going through the hidden nodes, and in the end reaching the output nodes. The algorithm used to train the network is the backpropagation algorithm. As mentioned earlier, backpropagation is short for "backward propagation of errors". Given an error function and an artificial neural network, the backpropagation algorithm calculates the gradient of the error function with respect to the weights of the neural network [39].

**Table 4.** Final values chosen for the training phase.

| Hyperparameter | Value |
|---|---|
| Batch Size | 64 |
| Epochs | 100 |
| Optimizer | Stochastic Gradient Descent (SGD) with Nesterov momentum |
| Momentum | 0.9 |
| Learning rate | 0.01 |
| Regularization | $1 \times 10^{-6}$ |

Moreover, the number of layers in the network is six: one input layer, one output layer and four hidden layers. The input layer takes the input dimension which is equal to the number of features used in the training dataset. The first hidden layer uses the Parametric Rectified Linear Unit (PReLU) activation function and it has 496 neurons. The PReLU activation function generalizes the traditional rectified unit with a slope for negative values and it is formally defined as [40]:

$$f(y_i) = \begin{cases} y_i & \text{if } y_i > 0 \\ a_i y_i & \text{if } y_i \leq 0 \end{cases} \tag{1}$$

The other hidden layers use the Rectified Linear Unit (ReLU) activation function. This function was designed to overcome the vanishing gradient problem and it works in the way that it returns 0 for any negative input, but for a positive input, it returns the value of the input back. It can be defined as:

$$f(x) = max(0, x) \tag{2}$$

The second, third and fourth hidden layers have 248, 124 and 62 nodes, respectively. The output layer has a different activation function and a different number of neurons based on the type of classification which is being done. For binary classification, the output layer uses the sigmoid activation function and has only one neuron. The sigmoid function takes a value as the input, and outputs another value between 0 and 1. It can be defined as:

$$f(x) = \frac{1}{1 + e^{-x}} \tag{3}$$

On the other hand, for the multiclass classification, the output layer has the number of neurons which is equal to the number of the attack categories in the dataset, and the activation function which is used is the softmax function. This function converts a vector of K real values into a vector of K real values that sum to 1 [41]. It can be defined as:

$$f_i(\vec{x}) = \frac{e^{x_i}}{\sum_{j=1}^{J} e^{x_j}} \text{ for } i = 1, ..., J \tag{4}$$

Additionally, to prevent overfitting during the training phase, we implemented the dropout on all the hidden layers. Dropout is a regularization method that causes some of the neurons of a layer to be randomly dropped out (ignored) during the training of the network. Dropping out the neurons means that they will not be considered during the specific forward or backward passing through the neural network. The dropout rate chosen for this network, for each hidden layer, was equal to 0.1. This means that 10% of the units will be dropped (set to 0) at each step. The units that are not dropped are scaled up by $\frac{1}{(1 - rate)}$ so that the sum of all the units remains unchanged. A graphical representation of the architecture of the neural network can be seen in Figure 3.

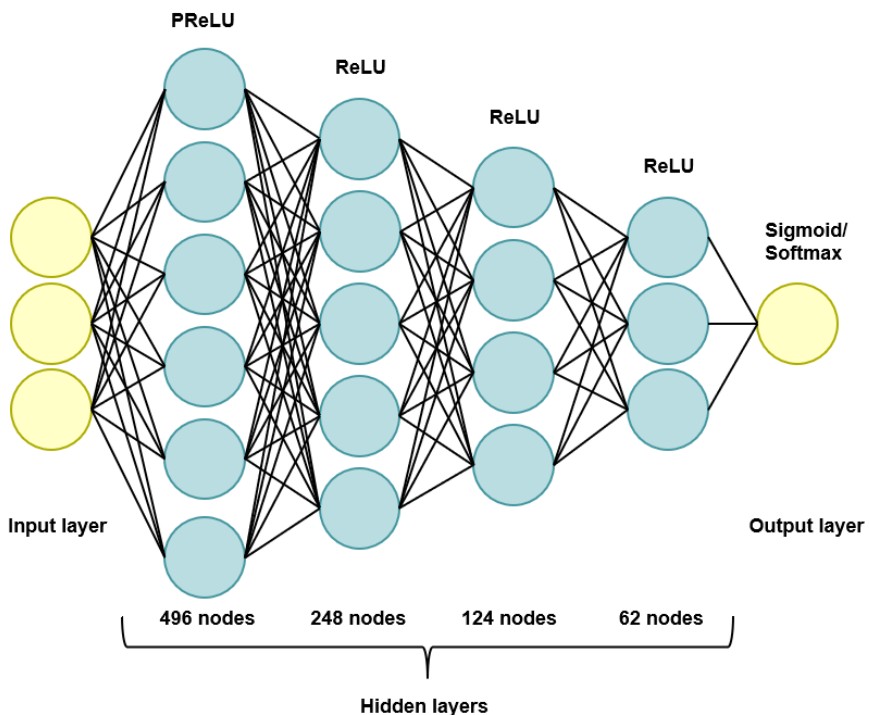

**Figure 3.** A graphical representation of the DNN architecture.

*4.4. Development Tools*

The data preparation, model implementation, training, testing, and evaluation were all done in *Python* using the following libraries:

- NumPy. This is a *Python* library which provides support for working with large multi-dimensional arrays. It allows the user to perform different mathematical operations on such arrays and it guarantees efficient calculations [42].
- Pandas. Pandas is a *Python* library used for data analysis and manipulation. It provides support for manipulating numerical tables and time series [43].
- Matplotlib. This is a *Python* library that provides support for data visualization. It is used to create static, animated and interactive graphs and other visualizations [44].
- Scikit-learn. This is a machine learning *Python* library used for predictive analysis. It is built on *NumPy, SciPy* and *Matplotlib* and it can provide features for classification, regression, model selection, clustering, preprocessing and so on. Another name for it is sklearn [45].
- Tensorflow. This is a *Python* library for machine learning. It provides features for building and training machine learning models and it allows users to create large scale neural networks with many layers [46].
- Keras. This is a *Python* library which provides an interface for artificial neural networks. It is built on top of *Tensorflow* and it acts as a frontend for it [47].
- Jupyter notebook. This is an interactive computational environment which allows the user to edit the code live, create equations, visualizations, and much more. It is practical for research because it allows the researcher to combine code, output, explanations, and multimedia resources in one document [48].
- PyCaret. This is an open-source *Python* library used for automation of the machine learning processes. It gives the user many options which include automatic data preparation, automatic model construction, training of the models, and evaluation and comparison of the models [49]. For this experiment, *PyCaret* was used to automate the data preparation and feature selection process.

All the experiments were conducted on a HP Pavilion Power laptop with the Intel(R) Core(TM) i7-7700HQ CPU @ 2.80 GHz processor. The rest of the hardware specifications of the laptop used for the experiment can be seen in Table 5.

**Table 5.** Hardware specifications of the computer used for training.

| Hardware | Specification |
| --- | --- |
| GPU | NVIDIA GeForce GTX 1050 |
| Memory | 16 GB system memory |
| Storage | 256 GB SSD |
| GPU Memory | 4 GB GPU memory |

## 5. Experiments

Our experimental campaign aimed to achieve the lowest False Negative Rate (FNR) while keeping the False Positive Rate (FPR) low. When it comes to multiclass classifications, an additional goal was to improve the accuracy of some of the classes which have a smaller number of records. The purpose of the experiments has been to find which architecture and hyperparameters give us the lowest FNR. Additionally, other performance metrics mentioned in Section 1 were compared for each experiment. A total of 14 experiments were conducted: 4 for binary classification on the NSL-KDD dataset, 4 for multiclass classification on the NSL-KDD dataset, three for binary classification on the UNSW-NB15 dataset, and three experiments for multiclass classification on the UNSW-NB15 dataset.

### 5.1. Experiments on the NSL-KDD Dataset

Since both binary and multiclass classification were done on this dataset, the first part of the experiments which will be explained were conducted for binary classification, and the second part for the multiclass classification.

#### 5.1.1. NSL-KDD Binary—Full Features

The first binary classification experiment considered the training of the model with all the features extracted during the data preparation phase (Section 4.2). Since the NSL-KDD subsets used for training and testing ($train_f$ and $test_f$) had a total of 123 columns each, the neural network's input layer has 123 nodes.

The first step is to train the neural network on this version of the training subset and assess the results achieved. We used the *Keras* library to build the model and fine-tune the hyperparameters, as mentioned in Section 4.3. We used Early Stopping (ES) to prevent overfitting the network. One problem which can lead to overfitting is using too many epochs to train the network. Hence, ES allows the user to set many training epochs, but it stops the process once the model performance reaches the best possible result and before it drops. The confusion matrix for this model can be seen in Table 6. The confusion matrix makes it easier to see which classes are easily confused by the model, and from this matrix it can be seen that the number of False Negatives (FN) is 3296, which is very high. This means that the model wrongly classified 3296 attack records as normal traffic. The number of false positives is equal to 662, which means that the model wrongly classified 662 records as attacks.

**Table 6.** Confusion matrix for NSL-KDD "full features" binary classification experiment.

| | | Predicted | |
| --- | --- | --- | --- |
| | | **Normal** | **Attack** |
| **Actual** | Normal | 9049 | 662 |
| | Attack | 3296 | 9537 |

### 5.1.2. NSL-KDD Binary—Modified Distribution

The second binary classification experiment considered using the same 123 features of the data preparation stage but slightly changing the training and testing subsets distribution. The idea behind this was that, maybe, the neural network could not learn from the training subset prepared by the authors. This experiment aims to see if a different distribution of records in the training and the testing subsets will give better results.

To obtain the new subsets, we combined the two subsets together into one dataset, shuffling the data and then splitting them again so that 80% of the records are used for the training of the network, and 20% of the records are used for testing. The training subset contained 118,813 records and the testing subset 29,704 records. The architecture of the neural network was the same as for the first experiment, and again, Early Stopping was used. The confusion matrix for this experiment can be seen in Table 7. It can be seen that the number of false negatives in this experiment is equal to 32, which is significantly lower than in the "NSL-KDD binary—full features" experiment. The number of false positives is is 52, which is also lower when compared to the previous experiment.

**Table 7.** Confusion matrix for NSL-KDD "modified distribution" binary classification experiment.

| | | Predicted | |
| --- | --- | --- | --- |
| | | **Normal** | **Attack** |
| **Actual** | Normal | 15,371 | 52 |
| | Attack | 32 | 14,249 |

### 5.1.3. NSL-KDD Binary—Reduced Features

For the third binary classification experiment, we used feature selection to reduce the False Positive and the False Negative Rates. The testing and training subsets used for this experiment were the same ones which were used for the second experiment. The feature selection process was automatized by using the *Python* library *PyCaret*. This library makes feature selection on a dataset by combining several supervised feature selection methods to select a subset of features that contribute the most to the prediction of the target variable [50].

After the feature selection process, the total number of features selected as the most important was 41 out of 123. This means that the model for this experiment had 41 neurons in the input layer. The rest of the architecture remained unchanged, including the use of the Early Stopping method. The confusion matrix for this experiment can be seen in Table 8. The number of false negatives in this experiment is 147, which is slightly higher than in the previous experiment, but still significantly lower than in the first NSL-KDD binary experiment. The number of false positives is 215, which is higher than in the previous experiment, but again, lower than in the first experiment.

**Table 8.** Confusion matrix for NSL-KDD "reduced features" binary classification experiment.

| | | Predicted | |
| --- | --- | --- | --- |
| | | **Normal** | **Attack** |
| **Actual** | Normal | 15,185 | 215 |
| | Attack | 147 | 14,157 |

### 5.1.4. NSL-KDD Binary—Class Weights

The fourth binary classification experiment included the use of class weights. When dealing with an imbalanced dataset, assigning weights to different classes can help the model make more accurate predictions. For our research, we consider the false negatives

more dangerous than false positives. Hence, we needed a way to make the model penalize the false negatives by assigning different class weights. We assigned a weight of 1 for the normal class (which has the label 0) and 2 for the attack class (which has the label 1). Aside from assigning weights to the classes, this experiment uses the same hyperparameters as the first and the second. The input dimension is equal to the second experiment since we considered the same 41 features of the feature selection phase. We used early Stopping in this experiment as well. The confusion matrix for this experiment can be seen in Table 9. From the confusion matrix it can be seen that the number of false negatives is 115, which is slightly lower than in the previous experiment, but still higher than in the "NSL-KDD binary—modified distribution" experiment. The number of false positives is higher than in the previous experiment.

**Table 9.** Confusion matrix for NSL-KDD "class weights" binary classification experiment.

|  |  | Predicted | |
|---|---|---|---|
|  |  | **Normal** | **Attack** |
| **Actual** | Normal | 15,024 | 376 |
|  | Attack | 115 | 14,189 |

5.1.5. NSL-KDD Multiclass—Full Features

The first multiclass classification experiment included the usage as an input in the neural network of all 123 features produced in the data preparation phase. The initial training and testing subsets provided by the authors were used. After the division of the subsets into input and output subsets during the data preparation phase, $train_f$ and $test_f$ contain the 123 features and will be used as inputs in the training and testing of the network. As described in Section 4.2, $train_c$ and $test_c$ subsets, which contain the attack category, will be used as the output in the training and the testing phase. Hence, the neural network's input layer has 123 nodes, like the "NSL-KDD binary—full features" experiment. The output layer has five nodes, one for each of the four attack categories and an additional one for the records which represent regular traffic. As mentioned earlier, the loss function used for the multiclass classification is the sparse categorical cross-entropy function. We opted for this function since it is recommended when the output is made of integers. The other hyperparameters are the same as explained in Section 4.3. As for binary classification, we used Early Stopping to prevent the model from over-fitting. Table 10 shows the confusion matrix for this example. To calculate the False Negative Rate, the class 'Normal' will be considered as the negative class, and the others as the positive classes. By taking a look at the confusion matrix, it can be concluded that the last column of the matrix shows the classes which were predicted as the 'Normal' (negative) class, so in the intersection of the last column and the last row, we have the number of True Negatives (TN). The TN in this case are the records which actually belong to the 'Normal' class and were correctly classified as the 'Normal' class. The other elements of the last column are false negatives (FN), meaning that they are records which actually belong to other classes and were wrongly classified as the 'Normal' class. Furthermore, the other elements in the last row are false positives (FP) since they actually belong to the 'Normal' class but were wrongly classified as attacks. All the other elements can be considered as true positives (TP) in this case. Taking this into account, the False Negative Rate can be calculated using these values and it is equal to 31.17%, which is a very unsatisfactory value.

**Table 10.** Confusion matrix for NSL-KDD "full features" multiclass classification experiment.

| | | Predicted | | | | |
|---|---|---|---|---|---|---|
| | | **DoS** | **Probe** | **R2L** | **U2R** | **Normal** |
| **Actual** | DoS | 5836 | 223 | 71 | 0 | 1330 |
| | Probe | 266 | 1835 | 1 | 0 | 319 |
| | R2L | 105 | 289 | 148 | 2 | 2341 |
| | U2R | 2 | 25 | 22 | 8 | 10 |
| | Normal | 387 | 243 | 3 | 0 | 9078 |

### 5.1.6. NSL-KDD Multiclass—Modified Distribution

The second multiclass classification experiment was conducted using the same logic as for the second binary classification experiment. Again, all 123 features were used as the input in the neural network, therefore the input layer has 123 nodes. The two original subsets provided by the authors are mixed into one, shuffled, and split, to obtain a different distribution of the testing and training subsets. After the split, the training subset contains 80% of the records, while the testing subset contains 20%. After the split, the training subset contains 118.813 records and the testing subset contains 29.704 records. Again, the Early Stopping method was used. The confusion matrix for this experiment can be seen in Table 11. Using the same logic as in the previous experiment, for calculating the False Negative Rate, the "Normal" class will be considered as the negative class. The FNR in this experiment is equal to 0.17%. By looking at the confusion matrices in Tables 10 and 11, it can be seen that the classes R2L and U2R have less records in the testing subset than in the previous experiment. By having more records in the training subset, and less in the testing one, the network learned to better classify records belonging to these classes. In fact, the testing subset used for this experiment contains 716 records belonging to the R2L class, and 20 belonging to the U2R class, and in case of the "full features" experiment, the testing subset contained 2885 R2L and 64 U2R records.

**Table 11.** Confusion matrix for NSL-KDD "modified distribution" multiclass classification experiment.

| | | Predicted | | | | |
|---|---|---|---|---|---|---|
| | | **DoS** | **Probe** | **R2L** | **U2R** | **Normal** |
| **Actual** | DoS | 10,666 | 2 | 1 | 0 | 7 |
| | Probe | 4 | 2806 | 1 | 0 | 9 |
| | R2L | 0 | 2 | 706 | 5 | 6 |
| | U2R | 0 | 1 | 5 | 12 | 2 |
| | Normal | 3 | 4 | 25 | 0 | 15,437 |

### 5.1.7. NSL-KDD Multiclass—Reduced Features

This experiment considered the use of the same subsets generated in the previous experiment, preforming feature selection with *PyCaret* and then training the network by using only the selected features. Out of 123 features, only 35 features were selected for this model. The next step was to train the neural network by using these 35 selected features, which means that the input layer of the neural network in this case had 35 nodes, one for every feature used for training. The confusion matrix for this experiment is shown in Table 12. The FNR for this experiment was equal to 0.133%. By looking at Tables 11 and 12 it can be concluded that the "NSL-KDD multiclass—modified distribution" model achieves better performance for the minority classes. On the other hand, the "NSL-KDD multiclass—reduced features" model achieves a lower FNR.

**Table 12.** Confusion matrix for NSL-KDD "reduced features" multiclass classification experiment.

|        |        | Predicted |       |      |      |        |
|--------|--------|-----------|-------|------|------|--------|
|        |        | **DoS**   | **Probe** | **R2L** | **U2R** | **Normal** |
| Actual | DoS    | 10,666 | 3    | 1   | 0 | 6      |
|        | Probe  | 1      | 2803 | 8   | 1 | 7      |
|        | R2L    | 1      | 10   | 702 | 2 | 4      |
|        | U2R    | 0      | 0    | 9   | 9 | 2      |
|        | Normal | 8      | 26   | 62  | 0 | 15,373 |

5.1.8. NSL-KDD Multiclass—Class Weights

For this experiment, we used the training and testing subsets of the previous experiment and the 35 features selected using feature selection. In addition, we introduced the class weights. The goal of setting specific class weights, in this case, is to make the neural network learn to better differentiate between the classes with a smaller number of records (U2R and R2L), and that is done by giving those classes a higher weight. Moreover, by correctly classifying records that belong to those classes, the FNR should also be lowered. We resolved to use the *Scikit-learn* method `compute_class_weight` for computing the class weights. Because very few records belong to the U2R class, the weights returned by this function needed to be slightly altered to avoid overfitting and falsely classifying many records belonging to the U2R class. The final class weights used were: 0.55 for the DoS class, 2.11 for the Probe class, 7.51 for the R2L class, 24.03 for the U2R class, and 0.38 for the Normal class. Again, the network's input layer had 35 nodes for the 35 selected features. The confusion matrix is shown in Table 13. The FNR for this experiment was the lowest, and it was equal to 0.049%.

**Table 13.** Confusion matrix for NSL-KDD "class weights" multiclass classification experiment.

|        |        | Predicted |       |      |      |        |
|--------|--------|-----------|-------|------|------|--------|
|        |        | **DoS**   | **Probe** | **R2L** | **U2R** | **Normal** |
| Actual | DoS    | 10,629 | 19   | 23  | 0  | 5      |
|        | Probe  | 1      | 2799 | 13  | 6  | 1      |
|        | R2L    | 0      | 5    | 706 | 7  | 1      |
|        | U2R    | 0      | 0    | 6   | 14 | 0      |
|        | Normal | 27     | 76   | 88  | 0  | 15,278 |

*5.2. Experiments on the UNSW-NB15 Dataset*

As for the NSL-KDD dataset, for this dataset we built binary and multiclass classification models.

5.2.1. UNSW-NB15 Binary—Full Features

The first binary classification experiment included using all 200 features obtained during the data preparation phase by splitting the original dataset. The neural network architecture, and the training hyperparameters, are explained in Section 4.3. The neural network's input layer has 200 nodes, one for each feature used. As for the NSL-KDD experiments, we used Early Stopping for the UNSW-NB15 experiments, to prevent the neural network from overfitting. The confusion matrix is shown in Table 14. The number of false negatives is equal to 1662 and the number of false positives is 2965.

**Table 14.** Confusion matrix for UNSW-NB15 "full features" binary classification experiment.

|  |  | Predicted | |
| --- | --- | --- | --- |
|  |  | **Normal** | **Attack** |
| Actual | Normal | 389,003 | 2965 |
|  | Attack | 1662 | 18,254 |

### 5.2.2. UNSW-NB15 Binary—Reduced Features

The second binary classification experiment considered using a minimized set of features obtained using a combination of several feature selection methods implemented by *PyCaret*. The feature selection picked 53 features out of 200, which were labeled as the most relevant for the classification process. Hence, the neural network's input layer consisted of 53 nodes, while the training hyperparameters were the same as in the other experiments. The rest of the network architecture was the same as the one presented in Section 4.3. The whole process included, again, the Early Stopping mechanism. The confusion matrix is shown in Table 15. The number of false negatives is equal to 1431 and the number of false positives is 3257. The number of FN is slightly lower than in the previous experiment, while the number of FP slightly increased.

**Table 15.** Confusion matrix for UNSW-NB15 "reduced features" binary classification experiment.

|  |  | Predicted | |
| --- | --- | --- | --- |
|  |  | **Normal** | **Attack** |
| Actual | Normal | 388,177 | 3257 |
|  | Attack | 1431 | 18,485 |

### 5.2.3. UNSW-NB15 Binary—Class Weights

In this experiment, we incorporated the class weights into the network of the previous experiment. We used the same subsets for training and testing and the 53 features selected during the feature selection process. The training hyperparameters and the rest of the network architecture are the same as in the previous experiment. The class weights were assigned in the following manner: 1 for the normal class (labeled with 0) and 3 for the attack class (labeled with 1). The confusion matrix is shown in Table 16. The number of false negatives is equal to 56, which is the lowest value in all three binary experiments. The number of false positives increased, and is equal to 5536. The increase was expected, because there is a trade-off between the false positives and false negatives.

**Table 16.** Confusion matrix for UNSW-NB15 "class weights" binary classification experiment.

|  |  | Predicted | |
| --- | --- | --- | --- |
|  |  | **Normal** | **Attack** |
| Actual | Normal | 386,432 | 5536 |
|  | Attack | 56 | 19,860 |

### 5.2.4. UNSW-NB15 Multiclass—Full Features

The first multiclass classification experiment included the usage of all 200 features obtained during the data preparation process, as well as the subsets generated by splitting the main dataset. The input layer has 200 nodes, one for each feature, and the output layer has ten nodes, one for each possible class representing normal traffic and the other nine for each attack category. The loss function used for this model is the sparse categorical cross-entropy function, and the activation function in the output layer is the softmax function.

Early Stopping was used. The confusion matrix can be seen in Table 17. The confusion matrix shows that the model was not able to correctly predict any of the attacks that belong to the class 'Worms'. The reason for this is the fact that the dataset is very unbalanced, and there were only 38 records belonging to this class in the testing dataset. Other classes, besides the class 'Worms' which have a low number of records are: Shellcode, Backdoor, and Analysis. Considering the 'Normal' class as the negative class, looking at the Table 17, the number of true negatives can be found in the intersection of the 7th row and the 7th column, and it is equal to 390,912. The false positives are all the records that belong to the 'Normal' class, but were wrongly classified as an attack, and they can be seen in the 7th row. The false negatives are the records that represent an attack, but were wrongly classified as belonging to the 'Normal' class, and they can be seen in the 7th column. Based on this, the FNR can be calculated, and it is equal to 16.48%.

**Table 17.** Confusion matrix for UNSW-NB15 "full features" multiclass classification experiment.

| | | Predicted | | | | | | | | |
|---|---|---|---|---|---|---|---|---|---|---|
| | | Analysis | Backdoor | DoS | Exploits | Fuzzers | Generic | Normal | Recconnaissanse | Shellcode | Worms |
| Actual | Analysis | 7 | 20 | 4 | 25 | 310 | 0 | 89 | 0 | 0 | 0 |
| | Backdoor | 8 | 11 | 10 | 49 | 261 | 3 | 11 | 41 | 0 | 0 |
| | DoS | 9 | 8 | 28 | 496 | 359 | 53 | 62 | 62 | 15 | 0 |
| | Exploits | 9 | 21 | 21 | 4232 | 459 | 70 | 345 | 427 | 12 | 0 |
| | Fuzzers | 11 | 20 | 2 | 127 | 1560 | 20 | 2536 | 123 | 12 | 0 |
| | Generic | 0 | 0 | 6 | 406 | 356 | 4175 | 52 | 38 | 5 | 0 |
| | Normal | 0 | 0 | 0 | 190 | 540 | 11 | 390,912 | 180 | 13 | 0 |
| | Recconnaissanse | 0 | 0 | 1 | 102 | 309 | 4 | 172 | 2122 | 0 | 0 |
| | Shellcode | 0 | 0 | 0 | 22 | 20 | 4 | 34 | 140 | 84 | 0 |
| | Worms | 0 | 0 | 0 | 27 | 3 | 5 | 2 | 1 | 0 | 0 |

### 5.2.5. UNSW-NB15 Multiclass—Reduced Features

For this experiment, we used the same data subsets as the previous one and included a feature selection process to use only the most relevant features for the classification of the records in each of the ten categories. By using the feature selection method from *PyCaret*, 44 features were selected as the most important. The input layer has 44 nodes, one for each of the selected features. Early Stopping was used. The confusion matrix is shown in Table 18. In comparison to the results obtained in the previous experiment, in this experiment, the model had worse performance when it comes to correctly classifying the minority classes. None of the records belonging to the 'Analysis', 'Backdoor' and 'Worms' classes were correctly classified. However, the FNR for this experiment was equal to 12.69%, which was a bit lower than in the "UNSW-NB15 multiclass—full features" experiment.

**Table 18.** Confusion matrix for UNSW-NB15 "reduced features" multiclass classification experiment.

| | | Predicted | | | | | | | | |
|---|---|---|---|---|---|---|---|---|---|---|
| | | Analysis | Backdoor | DoS | Exploits | Fuzzers | Generic | Normal | Recconnaissanse | Shellcode | Worms |
| Actual | Analysis | 0 | 0 | 15 | 167 | 131 | 0 | 82 | 60 | 0 | 0 |
| | Backdoor | 0 | 0 | 19 | 138 | 116 | 6 | 16 | 99 | 0 | 0 |
| | DoS | 0 | 0 | 28 | 625 | 144 | 62 | 42 | 173 | 18 | 0 |
| | Exploits | 0 | 0 | 28 | 4218 | 212 | 111 | 163 | 849 | 15 | 0 |
| | Fuzzers | 0 | 0 | 14 | 413 | 1346 | 63 | 2101 | 456 | 18 | 0 |
| | Generic | 0 | 0 | 8 | 518 | 164 | 4186 | 31 | 125 | 6 | 0 |
| | Normal | 0 | 0 | 3 | 393 | 582 | 47 | 390,217 | 586 | 18 | 0 |
| | Recconnaissanse | 0 | 0 | 5 | 149 | 129 | 8 | 91 | 2328 | 0 | 0 |
| | Shellcode | 0 | 0 | 0 | 13 | 17 | 5 | 17 | 178 | 74 | 0 |
| | Worms | 0 | 0 | 0 | 25 | 0 | 7 | 0 | 5 | 1 | 0 |

5.2.6. UNSW-NB15 Multiclass—Class Weights

This experiment included the usage of the same subsets as for the previous experiment (44 selected features) but with the addition of the class weights. The weights were calculated using the function `compute_class_weight` from the *Python* library *Scikit-learn*. We further refined the obtained weights to avoid over-fitting. The final weights used for training were the following: 9 for class 0 (no-attack), 10 for class 1, 5 for class 2, 3 for class 3, 3 for class 4, 3 for class 5, 1 for class 6, 4 for class 7, 15 for class 8 and 20 for class 9. The neural network architecture was the same as in the second experiment, and Early Stopping was used. The confusion matrix can be seen in Table 19. When compared to the first two experiments, the "UNSW-NB15 multiclass—class weights" experiment has seen an improvement in the performance metrics for these classes. The FNR is equal to 0.77%, which is the lowest of all three experiments.

**Table 19.** Confusion matrix for UNSW-NB15 "class weights" multiclass classification experiment.

| | | Predicted | | | | | | | | |
|---|---|---|---|---|---|---|---|---|---|---|---|
| | | Analysis | Backdoor | DoS | Exploits | Fuzzers | Generic | Normal | Recconnaissanse | Shellcode | Worms |
| Actual | Analysis | 14 | 349 | 0 | 23 | 0 | 0 | 69 | 0 | 0 | 0 |
| | Backdoor | 0 | 322 | 0 | 11 | 9 | 0 | 3 | 46 | 0 | 3 |
| | DoS | 0 | 376 | 30 | 456 | 45 | 52 | 12 | 92 | 28 | 1 |
| | Exploits | 8 | 442 | 13 | 3964 | 209 | 73 | 53 | 682 | 142 | 10 |
| | Fuzzers | 0 | 387 | 2 | 202 | 3076 | 103 | 8 | 443 | 190 | 0 |
| | Generic | 1 | 339 | 1 | 390 | 41 | 4154 | 6 | 59 | 32 | 15 |
| | Normal | 38 | 11 | 6 | 382 | 3886 | 78 | 386,630 | 684 | 131 | 0 |
| | Recconnaissanse | 0 | 323 | 2 | 12 | 101 | 6 | 4 | 2237 | 25 | 0 |
| | Shellcode | 0 | 0 | 0 | 1 | 21 | 4 | 0 | 142 | 136 | 0 |
| | Worms | 0 | 0 | 0 | 26 | 0 | 1 | 0 | 5 | 1 | 5 |

**6. Results**

This Section provides a detailed explanation of the results which were obtained from the experimental campaign, with a comparison of the results.

*6.1. Results of the NSL-KDD Experiments*

The results which were obtained in the 4 binary experiments done on the NSL-KDD dataset can be seen in Table 20.

**Table 20.** Comparison of the results achieved in the NSL-KDD binary classification experiments.

| Experiment | Training Accuracy | Prediction Accuracy | Precision | Recall | F1 Score | ROC AUC Score | FPR | FNR |
|---|---|---|---|---|---|---|---|---|
| Full features | 99.77% | 82.44% | 93.51% | 74.32% | 82.82% | 83.75% | 6.82% | 25.68% |
| Modified distribution | 99.76% | 99.72% | 99.64% | 99.78% | 99.71% | 99.72% | 0.33% | 0.22% |
| Reduced features | 98.94% | 98.78% | 98.5% | 98.97% | 98.74% | 98.79% | 1.4% | 1.03% |
| Class weights | 98.32% | 98.35% | 97.42% | 99.2% | 98.3% | 98.38% | 2.44% | 0.8% |

Observing the table, we can see that the "NSL-KDD binary—modified distribution" experiment achieved the best results, with the lowest FPR and FNR. The "NSL-KDD binary—full features" experiment achieved the lowest results, which could mean that the initial training and testing subsets distribution was not appropriate. The "NSL-KDD binary—reduced features" and the "NSL-KDD binary—class weights" experiments achieved more or less similar results, with the fourth one having a slightly lower FNR, which was the goal. On the other hand, the FPR in the "NSL-KDD binary—class weights" experiment was higher than in the "NSL-KDD binary—reduced features" one, which was expected because there is a trade-off between the false positives and false negatives.

Table 21 reports the results obtained in the 4 multiclass experiments with the NSL-KDD dataset.

**Table 21.** Comparison of the results achieved in the NSL-KDD multiclass classification experiments.

| Experiment | Training Accuracy | Prediction Accuracy | Precision | Recall | F1 Score | ROC AUC Score | FNR |
|---|---|---|---|---|---|---|---|
| Full features | 99.82% | 74.99% | 74.68% | 74.99% | 70.89% | 93.55% | 31.17% |
| Modified distribution | 99.82% | 99.74% | 99.74% | 99.74% | 99.74% | 99.9% | 0.17% |
| Reduced features | 99.59% | 99.49% | 99.51% | 99.49% | 99.49% | 99.99% | 0.133% |
| Class weights | 99.09% | 99.06% | 99.14% | 99.06% | 99.09% | 99.97% | 0.049% |

Observing the results in Table 21, we can see that the lowest FNR was reached in the "NSL-KDD multiclass—class weights" experiment and the highest in the "NSL-KDD multiclass—full features" experiment. In fact, all the evaluation metrics from the "NSL-KDD multiclass—full features" experiment show very poor performance, which again might mean that the datasets which were pre-made by the authors need a feature selection when facing the FNR minimization problem. The "NSL-KDD multiclass—modified distribution" experiment has slightly higher precision, recall, and F1 score than the "NSL-KDD multiclass—reduced features" and the "NSL-KDD multiclass—class weights" experiments. Overall, the "NSL-KDD multiclass—modified distribution", "NSL-KDD multiclass—reduced features", and "NSL-KDD multiclass—class weights" experiments all have performance metrics that are >99%, and that can be considered a satisfactory result. When it comes to the specific performance of the classes with a lower number of records, the U2R and R2L classes, the best performance for them was achieved in the "NSL-KDD multiclass—modified distribution" experiment.

As mentioned earlier, another one of the goals of this research is to increase the detection rates of some specific classes which were shown to have low detection rates in previous works by other authors, as shown in Section 3. For this dataset, the classes that had the lowest detection rates were R2L and U2R, so we report the following performance

metrics specifically for these two classes: precision, recall, and F1 score. For the NSL-KDD multiclass experiments, the detailed results of those metrics are in Table 22. The column "No. of records" refers to the number of records belonging to those classes in the testing dataset.

**Table 22.** Performance metrics for U2R and R2L classes in NSL-KDD multiclass experiments.

| Experiment | Class | Precision | Recall | F1 Score | No. of Records |
|---|---|---|---|---|---|
| Full features | R2L | 60% | 5% | 9% | 2885 |
| | U2R | 80% | 12% | 21% | 67 |
| Modified distribution | R2L | 96% | 98% | 97% | 719 |
| | U2R | 71% | 60% | 65% | 20 |
| Reduced features | R2L | 90% | 98% | 94% | 719 |
| | U2R | 75% | 45% | 56% | 20 |
| Class weights | R2L | 84% | 98% | 91% | 719 |
| | U2R | 52% | 70% | 60% | 20 |

When it comes to the specific performance of the classes with a lower number of records, the U2R and R2L class, the best performance for them was achieved in the "NSL-KDD multiclass—modified distribution" experiment. Since Early Stopping was used in order to prevent the model from overfitting, the average number of epochs needed to reach the optimal result while training the model on the NSL-KDD dataset was 25. The average time needed to train the network for this dataset was approximately 3 min for each experiment.

*6.2. Results of the UNSW-NB15 Experiments*

The results achieved in the three binary classification experiments with the UNSW-NB15 dataset are in Table 23. We can observe that the prediction accuracy is very similar in all three experiments. However, there is a considerable variation between the precision and recall, especially between the "UNSW-NB15 binary—full features" and the "UNSW-NB15 binary—class weights" experiments. The observed reduction is likely because the "UNSW-NB15 binary—class weights" experiment produced more false positives and fewer false negatives. After all, there is a trade-off between those two when using class weights. The goal was to lower the FNR as much as possible, and the model used in the "UNSW-NB15 binary—class weights" experiment was the most successful.

**Table 23.** Comparison of the results achieved in the UNSW-NB15 binary classification experiments.

| Experiment | Training Accuracy | Prediction Accuracy | Precision | Recall | F1 Score | ROC AUC Score | FPR | FNR |
|---|---|---|---|---|---|---|---|---|
| Full features | 98.87% | 98.88% | 86.03% | 91.65% | 88.75% | 95.45% | 0.76% | 8.34% |
| Reduced features | 98.86% | 98.86% | 85.02% | 92.81% | 88.75% | 96% | 0.83% | 7.18% |
| Class weights | 98.63% | 98.64% | 78.2% | 99.72% | 87.66% | 99.15% | 1.41% | 0.28% |

Three multiclass classification experiment results for the UNSW-NB15 dataset are in Table 24. We can observe that the lowest FNR was achieved in the "UNSW-NB15 multiclass—class weights" experiment, jointly with the best performance for minority classes. The same experiment also reached the highest precision. However, it is noticeable as there was not a significant variation between the other metrics among all the performed experiments. Since the dataset has a minimal number of records representing attacks

compared to records representing regular traffic, even introducing the class weights, it is hard for the network to learn how to distinguish between the different classes since there are too few samples.

**Table 24.** Comparison of the results achieved in the UNSW-NB15 multiclass classification experiments.

| Exp. Name | Training Accuracy | Prediction Accuracy | Precision | Recall | F1 Score | ROC AUC Score | FNR |
|---|---|---|---|---|---|---|---|
| Full features | 97.91% | 97.87% | 97.55% | 97.87% | 97.65% | 99.87% | 16.48% |
| Reduced features | 97.91% | 97.7% | 97.44% | 97.7% | 97.48% | 99.85% | 12.69% |
| Class weights | 97.25% | 97.25% | 98.2% | 97.25% | 97.55% | 99.83% | 0.77% |

The dataset's lowest number of records classes are Worms, Shellcode, Backdoor, and Analysis. The goal is to try to increase the prediction capability for these classes, so we report in Table 25 the class-specific precision, recall, and F1 score for these four classes. As for the other dataset, the column "No. of records" refers to the number of records belonging to those classes in the testing dataset.

**Table 25.** Performance metrics for minority classes in UNSW-NB15 multiclass experiments.

| Experiment | Class | Precision | Recall | F1 Score | No. of Records |
|---|---|---|---|---|---|
| Full features | Analysis | 16% | 2% | 3% | 455 |
| | Backdoor | 14% | 3% | 5% | 394 |
| | Shellcode | 60% | 28% | 38% | 304 |
| | Worms | 0% | 0% | 0% | 38 |
| Reduced features | Analysis | 0% | 0% | 0% | 455 |
| | Backdoor | 0% | 0% | 0% | 394 |
| | Shellcode | 49% | 24% | 33% | 304 |
| | Worms | 0% | 0% | 0% | 38 |
| Class weights | Analysis | 23% | 3% | 5% | 455 |
| | Backdoor | 13% | 82% | 22% | 394 |
| | Shellcode | 20% | 45% | 28% | 304 |
| | Worms | 15% | 13% | 14% | 38 |

When compared to the first two experiments, the "UNSW-NB15 multiclass—class weights" experiment has seen an improvement in the performance metrics for all these classes. Since Early Stopping was used in order to prevent the model from overfitting, the average number of epochs needed to reach the optimal result while training the model on the UNSW-NB15 dataset was 20. The average time needed to train the network for this dataset was approximately 30 min for each UNSW-NB15 experiment.

## 7. Conclusions

This research focused on building a deep neural network and training it on two modern datasets for binary and multiclass classification. Despite other works in the literature, the primary goals of our research were to lower the False Negative Rate as much as possible while still keeping the False Positive Rate low and increasing the detection rate of minority classes (classes with a low number of records). We proposed a strategy made of three points: correction of the training and testing subset distribution, feature selection, and usage of

class weights. We ran an experimental campaign for two well-established datasets to verify the effectiveness of our strategy in lowering the FNR and increasing the performance of minority classes. In almost all of the experiments, a combination of feature selection and the assignment of correct class weights during the training phase of the neural network gave the best results in lowering the FNR. We observed that the assignment of the class weights needs to be used with caution since it can easily lead to over-fitting and an increase in the FPR. Even when used correctly, it will still give a slight increase in the FPR, as seen from the experiments in this research, but the number is still considered low enough. Only in the case of binary classification for the NSL-KDD dataset the usage of class weights was not the best method for achieving the lowest FNR. A more effective correction was modifying the distribution of the train and test subsets. Regarding multiclass classification, feature selection with class weights proved to be the best method to increase the performance of the minority classes.

Compared to the work surveyed in Section 3, the neural network models constructed in our research incidentally outperform all of them in terms of accuracy, except [35]. This comparison can be seen in the overview given in Table 3. In terms of accuracy, our model reached accuracy values >99% for the NSL-KDD dataset, which is higher than the accuracy achieved by other models on the same dataset, both for binary and multiclass classification. When it comes to the results achieved for the UNSW-NB15 dataset, our proposed model reached the accuracy of >98% for binary classification, and >97% for multiclass classification. Once again, it has overcome most of the other models on the same dataset when it comes to accuracy, with the exception of [35], in which the model has achieved the accuracy of 98.99% for binary classification and the FPR of 0.56%. However, because the datasets used in this research are unbalanced, accuracy is not the best metric to evaluate the performance. Therefore, this research also uses precision, recall, F1 score, and ROC AUC score to assess the performance. The best results for the NSL-KDD dataset show that all of these metrics were >99%, and for the UNSW-NB15, they were >98% for binary classification and >97% for multiclass classification. When it comes to the FPR and FNR, when compared to the models surveyed in Section 3 where the authors focused on calculating these values, the values achieved by our proposed model, once again, outperform most of the surveyed models. The exception is once again [35], when it comes to binary classification for UNSW-NB15 dataset specifically.

The main limitation of our work is that we have limited evidence of the generalization of our strategy. This is because we used only two of the most established datasets to validate our approach. A natural extension of our experiment to other datasets would further confirm the validity of our approach.

We acknowledge that our results for the lowest-represented attack classes are not optimal, and there is still space for increasing the performance. However, the major problem remains: the number of their records is still too low for a deep neural network to learn from it. One of the possible directions could be finding a way to improve these datasets to fix the imbalance and therefore increase the detection rates for minority classes. One idea is to generate and add more records to the minority classes. Another alternative is to use oversampling techniques. Most of the hyperparameter tuning in this research was done manually by doing different experiments. One possible alternative we should consider in future research would be using automatic parameter tuning methods to achieve better performance. Another direction for future work would be to test these models in a live system with actual attacks to see how well they would perform in the real world.

**Author Contributions:** Conceptualization, J.M. and A.S.; methodology, J.M.; software, J.M.; validation, J.M. and A.S.; formal analysis, J.M.; investigation, J.M.; resources, J.M. and A.S.; data curation, J.M.; writing—original draft preparation, J.M. and A.S.; writing—review and editing, J.M. and A.S.; visualization, J.M.; supervision, A.S.; project administration, J.M. and A.S. All authors have read and agreed to the published version of the manuscript.

**Funding:** This work was supported in part by the MIUR under grant "Dipartimenti di eccellenza 2018-2022" of the Department of Computer Science of Sapienza University.

**Institutional Review Board Statement:** Not applicable.

**Informed Consent Statement:** Not applicable.

**Data Availability Statement:** Publicly available datasets were analyzed in this study. This data can be found here: https://www.unb.ca/cic/datasets/nsl.html and https://research.unsw.edu.au/projects/unsw-nb15-dataset (accessed on 4 June 2022).

**Conflicts of Interest:** The authors declare no conflict of interest.

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
