# Peer review of "Reducing the False Negative Rate in Deep Learning Based Network Intrusion Detection Systems"

_algorithms, doi:10.3390/a15080258_

Round 1
Reviewer 1 Report
The manuscript proposes a deep learning model for network intrusion detection to perform both binary and multiclass classification of network traffic with respect to attacks. The methodology is evaluated on two benchmark datasets (NSL-KDD and NSW-NB15).
Comments:
1. The novelty of the proposed approach is not clear. I see standard practices used in this study.
2. Figure 1 is trivial. I suggest to remove it.
3. Avoid mass citation practices such as [9] [11] [12] [13] [14] [15] [16] [17] [18], but rather discuss each reference separately.
4. The related works section discusses previous studies, but some of them are outdated and no longer represent the state-of-the-art in this rapidly evolving field of research. There is a lack of focus and structure. The selection of works to discuss seems to be ad hoc. I suggest to focus on the recent state-of-the-art published from 2020 onwards, such as for example * Threat analysis and distributed denial of service (DDoS) attack recognition in the internet of things (IoT). * A modified grey wolf optimization algorithm for an intrusion detection system. * A novel approach for network intrusion detection using multistage deep learning image recognition.
5. Present a workflow diagram of your approach. Figure 2 is just a collection of steps and it is not detailed enough.
6. I do not understand why for two problems (binary and multi-class classification) you needed to divide the datasets into three parts.
7. How do you arrive at the hyperparameter values presented in table 4. Did you use some hyperparameter optimisation/tuning methods? Present the results of the ablation study.
8. Analyze the computational complexity of your approach.
9. Present and discuss the confusion matrices.
10. Compare you results with the results of other authors achieved on the same benchmark datasets.
11. Discuss the limitations of the proposed methodology.
12. Improve the conclusions. Use the results from experiments to support your claims.
Author Response
We sincerely thank the anonymous reviewer for her valuable comments and suggestions. We have carefully considered and addressed each point and revised the whole manuscript. Hereafter, we detail each point singularly.
1. The novelty of the proposed approach is not clear. I see standard practices used
in this study.
We agree with the reviewer that the novelty elements of our research were not apparent in the previous version of the paper. For this reason, we revised the whole paper, also expanding both the abstract and introduction, stressing that the study aimed to address the problems of the False Negative rate and the detection of the lower represented attack categories that have not been adequately addressed in the literature.
2. Figure 1 is trivial. I suggest to remove it.
As suggested, we have removed the figure.
3. Avoid mass citation practices such as [9] [11] [12] [13] [14] [15] [16] [17] [18], but
rather discuss each reference separately.
The citation sequence enumerated some of the works we revised in the literature review section. We have removed the sequence and referred the discussion to the Related work section.
4. The related works section discusses previous studies, but some of them are
outdated and no longer represent the state-of-the-art in this rapidly evolving field of
research. There is a lack of focus and structure. The selection of works to discuss
seems to be ad hoc. I suggest to focus on the recent state-of-the-art published
from 2020 onwards, such as for example * Threat analysis and distributed denial
of service (DDoS) attack recognition in the internet of things (IoT). * A modified
grey wolf optimization algorithm for an intrusion detection system. * A novel
approach for network intrusion detection using multistage deep learning image
recognition.
We thank the reviewer for this helpful comment. We expanded and renewed the literature review, including the suggested references and several other recent works. We also included in the comparison table 4 a more recent work. The number of citations now is 51.
5. Present a workflow diagram of your approach. Figure 2 is just a collection of
steps and it is not detailed enough.
Thanks for raising this point. We have included a new figure representing our approach strategy and used the original Figure 1 as a reference for the overall model generation procedure we followed.
6. I do not understand why for two problems (binary and multi-class classification)
you needed to divide the datasets into three parts.
We explained better that the step is only needed to distinguish between the subsets used for the two different classifications. We think that the revised section clarifies the motivation of our step.
7. How do you arrive at the hyperparameter values presented in table 4. Did you
use some hyperparameter optimisation/tuning methods? Present the results of the ablation study.
We included the missing details in Section 4.3 that a manual tuning was performed.
8. Analyze the computational complexity of your approach.
We included the average time for each experiment in Sections 6.1 and 6.2, which are an average of 3 minutes and 30 minutes for the NSL-KDD and the UNSW-NB15 datasets, respectively.
9. Present and discuss the confusion matrices.
We included and discussed all the confusion matrices for all the experiments in Section 5.
10. Compare you results with the results of other authors achieved on the same
benchmark datasets.
We included the results of our experiments in Table 3, comparing our results with those analyzed in the related work section.
11. Discuss the limitations of the proposed methodology.
We detailed in the conclusions that the main limitation is the lack of evidence about the generalization of our strategy.
12. Improve the conclusions. Use the results from experiments to support your
claims.
We have completely revised the Conclusions section. We recalled the primary goals of our research and the details of our proposed strategy. Moreover, we discussed all the experiments' results to support our proposal. Finally, we have also reported the major limitation of our approach.
Reviewer 2 Report
Dear Authors:
Thank you for your effort in the manuscript. I would like to suggest some issues:
1- section "1.1. Contributions ", it is not stating a real contribution. I suggest you delete it. According to your manuscript you been testing/evaluating "most effective methods for lowering the FNR and increasing the predictability for the minority classes". This is your main contribution and you should make it clear at every location in the manuscript.
2- The contribution at point 1 should have simple diagram to explain it, or may be a pseudo code.
3- The deep learning model you used, make a figure to show how many layers in it.
4- I would assume that the following techniques is what you meant by "we evaluated several models to identify the most effective methods for lowering the FNR"
a) Modified distribution
b) Reduced features
c) Class weights
You can mention it from the beginning of the manuscript, also if it was developed by other authors, please cite them.
Wish you best of luck
Author Response
Thank you for your effort in the manuscript. I would like to suggest some issues:
We thank the anonymous reviewer for her appreciation and suggestions. In the following, we detail how we addressed each point, one by one.
1- section "1.1. Contributions ", it is not stating a real contribution. I suggest you
delete it. According to your manuscript you been testing/evaluating "most
effective methods for lowering the FNR and increasing the predictability for
the minority classes". This is your main contribution and you should make it clear
at every location in the manuscript.
Thank you very much for this helpful suggestion. We have revised the manuscript, emphasizing this point from the beginning, namely in the abstract, in the introduction, and in the following sections. We are confident that the point should be more explicit in this revised version.
2- The contribution at point 1 should have simple diagram to explain it, or may be a pseudo code.
This was an excellent suggestion: we have included both the diagram and the explanation of our proposed strategy in Section 1 and Section 4.
3- The deep learning model you used, make a figure to show how many layers in it.
We have included a figure with the network architecture in Section 4.3.
4- I would assume that the following techniques is what you meant by "we
evaluated several models to identify the most effective methods for lowering
the FNR"
a) Modified distribution
b) Reduced features
c) Class weights
You can mention it from the beginning of the manuscript, also if it was developed
by other authors, please cite them.
Thank you so much. We have reported our strategy according to this valuable suggestion.
Wish you best of luck
Again, thank you so much.
Reviewer 3 Report
The paper covers a topical issue, presented in a sufficient manner. However, there is a list of improvements to be made before the paper can be accepted for its publication.
Statements such as "One way of improving IDSs, on which the researchers have been working in the last 27 years, is using machine learning techniques to reduce the FPR and FNR and improve 28 general detection capabilities." should be supported with the evidence from the literature. As an example, see Azeroual, O., & Nikiforova, A. (2022). Apache spark and MLlib-based intrusion detection system or how the big data technologies can secure the data. Information, 13(2), 58.
Same is the case for "Another reason for using ML algorithms 31 is that they are not domain-dependent and are relatively manageable to be functional for 32 multiple problems", and "Researchers identified two main issues in the literature regarding the already existing 34 deep learning models used for IDS" and the following breakdown into these issues.
This is even more the case when the authors define what the IDS actually is. For this you could refer to the reference used in the above article, i.e. Khraisat, A.; Gondal, I.; Vamplew, P.; Kamruzzaman, J. Survey of intrusion detection systems: Techniques, datasets and challenges. Cybersecurity 2019, 2, 20 AND for the breakdown into the types of IDSs.
I would suggest to reconsider the need for splitting the Introduction in Subsections and putting them into the main text of the Introduction.
The authors are also invited to avoid some too formal statements such as "One nice aspect of this type of IDS", i.e. "nice". In addition, it is typically considered to be a so-called "empty word". This is also the case for example such as "If we do not give helpful input to the model" and "helpful" in particular, "... we could proceed with doing the rest of the data preparation on the subsets as they are". Look for more appropriate terms, which would be sufficiently scientifically soundy.
The list of references is highly insufficient for the journal article. The authors are invited to pay more attention to the scientific literature since most of 38 sources used so far are to the external websites, which are not the scientific sources. This puts under the question the scientific value (actually the related works were covered well with the reference etc. but the rest of the paper very poorly reflects on the literature even in the case of providing the definitions for the concepts). It is, but this should be proved by providing better support through the evidence from the literature, especially the most recent ones.
The language requires improvements, referring to both grammar, style and flow. Although it is acceptable and easy to understand, but the involvement of the native speaker would be very recommended. And probably the most important point is the need to avoid too formal tense in some cases.
Table 3 presents a very interesting and concise overview of the relevant studies, however, the authors should elaborate on how they were selected? What was the process of their selection? was this a SLR? or what was the method applied? (if any)
Section cannot start with the figure, which is referred then in the text. Please, revise.
Author Response

(The authors gave the same response as above.)

Round 2
Reviewer 1 Report
The manuscript was well revised and can be accepted for publication.
Reviewer 2 Report
The authors modified the contents based on the directions given in round 1.
Reviewer 3 Report
The authors made a lot of improvements, answering all comments received from reviewers in both the paper and the letter. I found answers sufficient, same as changes done int he paper. I believe that the paper was significantly improved to accept it for publishing with a potential interest for a broad audience.